# An age-stratified serosurvey against purified *Salmonella enterica* serovar Typhi antigens in the Lao People´s Democratic Republic

Lisa Hefele[1,2], Antony P. Black[2], Trinh Van Tan[3], Nguyen Tri Minh[4,5], Nguyen Duc Hoang[4,5], Siriphone Virachith[2], Claude P. Muller[1], Judith M. Hübschen[1], Paula Russell[6,7], Josefin Bartholdson Scott[6,7], Chau Nguyen Ngoc Minh[3], Tran Vu Thieu Nga[3], Stephen Baker[6,7] *

1 Department of Infection and Immunity, Luxembourg Institute of Health, Esch-sur-Alzette, Grand-Duchy of Luxembourg, 2 Lao-Lux Laboratory, Institut Pasteur du Laos, Vientiane, Lao People's Democratic Republic, 3 The Hospital for Tropical Diseases, Wellcome Trust Major Overseas Programme, Oxford University Clinical Research Unit, Ho Chi Minh City, Vietnam, 4 University of Natural Sciences, Ho Chi Minh City Vietnam, 5 Vietnam National University, Ho Chi Minh City, Vietnam, 6 Cambridge Institute of Therapeutic Immunology and Infectious Disease, University of Cambridge School of Clinical Medicine, Cambridge Biomedical Campus, Cambridge, United Kingdom, 7 Department of Medicine, University of Cambridge School of Clinical Medicine, Cambridge Biomedical Campus, Cambridge, United Kingdom

* sgb47@medschl.cam.ac.uk

**Data Availability Statement:** All relevant data are within the manuscript and its Supporting Information files.

## Abstract

The epidemiology of typhoid fever in Lao People's Democratic Republic is poorly defined. Estimating the burden of typhoid fever in endemic countries is complex due to the cost and limitations of population-based surveillance; serological approaches may be a more cost-effective alternative. ELISAs were performed on 937 serum samples (317 children and 620 adults) from across Lao PDR to measure IgG antibody titers against Vi polysaccharide and the experimental protein antigens, CdtB and HlyE. We measured the significance of the differences between antibody titers in adults and children and fitted models to assess the relationship between age and antibody titers. The median IgG titres of both anti-HylE and CdtB were significantly higher in children compared to adults (anti-HylE; 351.7 ELISA Units (EU) vs 198.1 EU, respectively; $p<0.0001$ and anti-CdtB; 52.6 vs 12.9 EU; $p<0.0001$). Conversely, the median anti-Vi IgG titer was significantly higher in adults than children (11.3 vs 3.0 U/ml; $p<0.0001$). A non-linear trend line fitted to the anti-CdtB and anti-HlyE IgG data identified a peak in antibody concentration in children <5 years of age. We identified elevated titers of anti-HlyE and anti-CdtB IgG in the serum of children residing in Lao PDR in comparison to adults. These antigens are associated with seroconversion after typhoid fever and may be a superior measure of disease burden than anti-Vi IgG. This approach is scalable and may be developed to assess the burden of typhoid fever in countries where the disease may be endemic, and evidence is required for the introduction of typhoid vaccines.

**Funding:** This work was supported by the Ministry of Foreign and European Affairs, Luxembourg and the Luxembourg Institute of Health (project "Luxembourg-Laos Partnership for Research and Capacity Building in Infectious Disease Surveillance II") to LH and APB and a Wellcome senior research fellowship to SB (215515/Z/19/Z). The funders had no role in the design and conduct of the study; collection, management, analysis, and interpretation of the data; preparation, review, or approval of the manuscript; and decision to submit the manuscript for publication.

**Competing interests:** The authors have declared that no competing interests exist.

## Author summary

Typhoid fever is a serious bloodstream infection caused by the bacterium *Salmonella* Typhi. Estimating the burden of typhoid fever is complex due to the limitations, cost, and scalability of current diagnostic surveillance methods. The detection of specific antibody responses against the organism may be a more sustainable manner of measuring exposure and disease burden in endemic location. We measured antibody (IgG) in 937 serum samples (317 children and 620 adults) from across the Lao People's Democratic Republic against a polysaccharide (Vi) and two experimental protein antigens, CdtB and HlyE, that may more appropriate markers of disease exposure. We measured the significance of the differences between antibody titers in adults and children and fitted models to assess the relationship between age and antibody titers. The median IgG titres against HylE and CdtB were significantly higher in children than adults. Conversely, the median IgG titres against Vi was significantly higher in adults than children. We identified a significant association between a peak in IgG titres against CdtB and HlyE in children aged under 5 years. These data are indicative of high level of typhoid fever exposure in children under 5 years of age in Lao PDR and we surmise that IgG titres against HylE and CdtB may be a superior measure of typhoid disease burden than IgG titres against Vi. Our approach is scalable and can be further validated to assess the burden of typhoid fever in countries where the disease may be endemic, and evidence is required for the introduction of typhoid vaccines.

## Introduction

Typhoid fever is a systemic disease caused by *Salmonella enterica* subspecies enterica serovar Typhi (*S.* Typhi), a bacterium transmitted via contaminated food or water. Globally, an estimated 128,000–161,000 people per year die as a consequence of this infection [1]. Typhoid fever is typically diagnosed clinically, with blood culture as confirmatory gold standard [2–5]. Additionally, despite its limited performance, the serological Widal test is still commonly used [3,6]. However, all current diagnostic tests for typhoid fever have limitations and new technologies are constantly being evaluated [7,8].

A lack of longitudinal incidence data in many countries where typhoid fever is suspected to be endemic is a major barrier for the introduction of typhoid conjugate vaccines (TCVs), as past or current typhoid fever disease burden represents the evidence base for vaccination policy[9]. Consequently, there is a need for new approaches that can assess the extent of typhoid fever infections without automated blood culture systems or expensive population-based studies. Serological markers to assess typhoid fever prevalence may be a suitable approach for generating disease estimates and accounting for subclinical infections. *S.* Typhi exposure in the general population is not routinely evaluated in cross-sectional studies, but serological assays have been used to measure seroprevalence [10–12].

Antibodies against hemolysin E (anti-HlyE) and cytolethal distending toxin subunit B homolog (anti-CdtB) have been identified as potential biomarkers for identifying typhoid fever cases/exposure [7,13–15]. HlyE and CdtB are expressed in *S.* Typhi and *S.* Paratyphi A, but uncommon in other *Salmonella* spp. [16]. Additionally, antibody responses against the capsular polysaccharide Vi antigen (anti-Vi), the major component of TCVs, have also been used to assess exposure [10,11,17]. The Vi antigen is only present in *S.* Typhi, *S.* Dublin and *S.* Paratyphi C, but is absent from *S.* Paratyphi A and most gastroenteritis-causing serovars[18].

Typhoid fever is a notifiable disease in the Lao People's Democratic Republic (PDR), and several outbreaks have been reported between 2012 and 2017 [19–21]. A study estimated that the annual incidence of typhoid fever in Vientiane was 4.7 per 100,000 persons between 2015 and 2017 [22]. More recently, a study conducted in Vientiane over 18 years reported that the annual number of typhoid fever patients decreased from 2010 onwards; the estimated annual incidence of typhoid fever was 0.59 per 100,000 people in 2018 [20]. Healthcare access is inadequate in Lao PDR, with blood culture capability limited to only three laboratories nationally [21]. Consequently, typhoid fever surveillance and confirmation is inadequate, which impacts on an accurate assessment of the disease burden, and ultimately hindered the introduction of TCV. There is a need for better methods to estimate the burden of typhoid fever in endemic countries; specifically, to identify locations and age groups that have the highest exposure to *S.* Typhi.

To provide more data on the circulation of *S.* Typhi in Lao PDR, we conducted a serological, cross-sectional study using serum samples from different age groups and geographical areas of Lao PDR. This study is the first serology-based study for typhoid fever in Lao PDR and provides initial insights into age-associated exposure to *S.* Typhi and baseline antibody titers against Vi and HlyE and CdtB in the general population.

## Methods

### Ethical statement

The studies generating samples and data for this study were approved by the Lao National Ethics Committee (Cohort 1: 059/2013/NECHR, 022/2014/NECHR, 033/2017/NECHR, 032/2017/NECHR, 031/2017/NECHR, 056/2017/NECHR, and 038/2016/NECHR; Cohort 2: NECHR 059/2013 and 059/2014). Formal written informed consent was taken from all individuals enrolled in the previous studies, in the case of those aged under 16 years this was provided by a parent or guardian.

### Study population

The samples for this study originated from independent child and adult cohorts. The child cohort was comprised of 317 children and adolescents aged between 9 months and 15 years. These individuals were selected randomly (with respect to age and sex) from participants who were recruited within the framework of three other studies [23,24]. The studies generating these serum samples were conducted in central Lao PDR between 2017 and 2018. Two of these studies were cross-sectional seroprevalence studies focusing on vaccine-preventable diseases [23,24]. The third study was a hospital-based study focusing on transfusion-transmissible infections in blood transfusion recipients; only the control samples from this study were subjected to anti-*Salmonella* ELISAs. The adult cohort was comprised of 620 blood donors aged between 17 and 40 years who were recruited in the context of another research study between 2013 and 2015. The samples were randomly selected from a total of 5,018 and stratified by age, sex, and province. Typhoid fever vaccination is not part of the national immunization schedule in the Lao PDR and it is unlikely that participants of these studies received a typhoid fever vaccine.

### Serological testing

To determine the concentration of anti-Vi IgG antibodies, a commercial ELISA kit (Vacczyme, Binding site, UK) was employed according to the manufacturer's instructions. Antibody

concentrations were derived from the optical density (OD) data using a standardized curve-fitting 4-parameter logistic method.

Anti-HlyE IgG and anti-CdtB IgG in-house ELISAs were performed according to a previously described protocol, both antigens were also purified in house [7]. Briefly, 96 well flat-bottom ELISA plates (NunC 442404, Thermo Scientific) were coated overnight with 100 µl per well of the various antigens (final concentrations; 7 µg/ml of CdtB antigen and 1 µg/ml of HlyE antigen in 50 mM Carbonate Bicarbonate buffer). Coated plates were washed and blocked with 5% milk solution in Phosphate-buffered saline for two hours. After the blocking, plates were washed and incubated with 100µl sample (1:200 dilution) at room temperature. Plates were incubated with 100µl per well of alkaline phosphatase conjugated anti-human IgG (Sigma) for one hour at room temperature. Plates were developed using p-Nitrophenyl phosphate (SigmaFAST N1891, Sigma Aldrich, UK) substrate for 60 minutes at ambient temperature and the final absorbance was read at dual wavelengths (405 nm and 490 nm) using an automated microplate reader. Antibody concentrations in ELISA units (EU) were derived from the OD data using a standardized curve-fitting 4-parameter logistic method. If the measured antibody concentration was above or below the calculation range, the sample was tested again in a higher or lower dilution.

### Data analysis

Anti-Vi IgG data containing left-censored values were analyzed using methods described in the NADA package [25]. The left-censored data were stored using an indicator variable: The first variable contained the measured titer data and values below the calculation limit were stored as the lowest limit (7.4 U/ml). The second variable indicated if the value was a true measurement or censored. Summary statistics of the anti-Vi IgG data were calculated using robust regression on order statistics to account for left-censoring of the data. Antibody titers measured by ELISA were log transformed. After analysis of normality of independent variables in the different groups (using the Shapiro-Wilk test) and homogeneity of the variances between the groups (using Levene's test), non-parametric statistical tests were employed. Wilcoxon test or Kruskal-Wallis test followed by Dunn's multiple comparison tests with Bonferroni correction were used to test the significance of the differences between the antibody titer measured in groups. In case of the left-censored anti-Vi IgG data, a generalized Wilcoxon test was used ("cendiff", NADA package [25]). The Spearman correlation coefficient or Kendall´s tau (in case of censored data) were calculated to measure the association between antibody levels determined by ELISA.

In order to assess the relationship between age and anti-HlyE IgG and anti-CdtB IgG antibody levels, both a linear regression model and a generalized additive model were fitted to the data. Generalized additive models are regression-based models that estimate non-linear trends for the predictor variable without making assumptions about the shape of the function. The anti-Vi IgG data was fitted as a function of age using Akritas–Theil–Sen non-parametric regression to account for the left-censored data. A $p$ value $<0.05$ was considered statistically significant. Data analyses were conducted using R [26] with tidyverse [27], ggbeeswarm [28], ggpubr [29], mgcv [30], rstatix [31], fitdistrplus [32], plotrix [33] and NADA [25].

## Results

### Population characteristics

In total, sera from 937 participants originating from a range of provinces across Lao PDR were included in the study (Table 1 and S1 Fig). The majority of children in the child cohort were from Vientiane (249/317; 78.6%) and most (171/317; 53.9%) were female. The age of the

**Table 1. Characteristics of study participants in adult and child cohorts.**

| | | Children (n = 317) | Adults (n = 620) |
|---|---|---|---|
| Study year | | 2017–2018 | 2013–2015 |
| Age (years) | min—max | 0–15 | 17–40 |
| | mean | 7.56 | 26.34 |
| | median | 8 | 26 |
| Sex (%) | male | 46.06 | 60.16 |
| | female | 53.94 | 39.84 |
| Province (%) | Attapeu | 0.00 | 13.23 |
| | Bolikhamxay | 21.45 | 0.00 |
| | Huaphan | 0.00 | 10.00 |
| | Khammouane | 0.00 | 17.42 |
| | Luang Namtha | 0.00 | 11.61 |
| | Luang Prabang | 0.00 | 16.13 |
| | Vientiane Province & Capital | 78.55 | 15.65 |
| | Phongsaly | 0.00 | 4.35 |
| | Xayabouli | 0.00 | 11.61 |
| Profession (%) | office worker | - | 28.06 |
| | soldier | - | 25.48 |
| | student | - | 37.42 |
| | other | - | 9.03 |

children ranged from 0 to 15 years, with a median age of 8 years. The majority (373/620; 60.2%) of the adult participants were male (Table 1) and over a third (232/620; 37.4%) were students. The age of the adult participants ranged from 17 to 40 years (median 26 years).

## The seroprevalence of anti–S. Typhi IgG antibodies

We measured IgG antibodies targeting HlyE, CdtB, and Vi antigen in serum from the 937 participants. Overall, the anti-Vi antibody titers ranged from 7.4 U/ml to 600 U/ml, the anti-HlyE IgG antibody titers ranged from 12.6 EU to 5163.2 EU, and the anti-CdtB IgG antibody titers ranged from 2.8 to 1466.1 EU (Table 2). Notably, 469/937 (50.1%) of the samples generated anti-Vi IgG titers that were below the calculation limit of 7.4 U/ml. The mean anti-HlyE, anti-CdtB, and anti-Vi IgG titers among all participants were 453.8 EU, 16.8 EU and 7.5 U/ml, respectively (Table 2).

The distribution of antibody responses to the various antigens in adults and children is shown in Fig 1. These data demonstrated a clear delineation between the distribution of antibody titers between children and adults. For example, there was a significant difference in median anti-HlyE IgG titer between children (351.7 EU) and adults (198.1 EU; $p<0.0001$, Wilcoxon test) (Fig 1A). Similarly, the median anti-CdtB IgG was significantly higher in children than adults (52.6 vs 12.9 EU; $p<0.0001$, Wilcoxon test) (Fig 1B). We also observed a difference between the anti-Vi IgG titers in children and adults; however, contrary to the protein antigens, the median anti-Vi IgG titer was significantly higher in adults than children (11.3 vs 3.0 U/ml; $p<0.0001$, Wilcoxon test) (Fig 1C, Table 2). Data from both children and adults was available from Vientiane. We observed the same difference between the distributions of antibody titers between children and adults in this subset of data (S2 Fig).

The anti-HlyE IgG and anti-CdtB IgG titers demonstrated a significant positive correlation with each other (Spearman´s rho = 0.5; $p<0.00001$) (Fig 2A). Notably, the correlation coefficient of anti-HlyE IgG and anti-CdtB IgG was substantially higher among children (Spearman

**Table 2. Anti–S. Typhi serum IgG antibody titers in the adult and child cohorts.**

| | | N | N cens | Median | Mean | sd | Max | Min |
|---|---|---|---|---|---|---|---|---|
| anti-HlyE IgG (EU) | All data | 937 | 0 | 234.32 | 453.83 | 672.37 | 5163.20 | 12.64 |
| | Cohort 1: Children | 317 | 0 | 351.74 | 734.59 | 980.75 | 5163.20 | 28.34 |
| | Cohort 2: Adults | 620 | 0 | 198.14 | 310.29 | 362.72 | 4520.40 | 12.64 |
| anti-CdtB IgG (EU) [1] | All data | 935 | 0 | 16.80 | 77.15 | 176.20 | 1466.10 | 2.75 |
| | Cohort 1: Children | 317 | 0 | 52.59 | 178.68 | 270.01 | 1466.10 | 3.73 |
| | Cohort 2: Adults | 618 | 0 | 12.87 | 25.07 | 40.57 | 470.18 | 2.75 |
| Anti-Vi IgG (U/ml) | | | | | | | | |
| All observations | All data | 937 | 469 | 7.53 | 27.84 | 59.01 | 600.00 | 7.40 |
| | Cohort 1: Children | 317 | 218 | 3.02 | 10.34 | 23.15 | 204.60 | 7.42 |
| | Cohort 2: Adults | 620 | 251 | 11.32 | 36.71 | 69.00 | 600.00 | 7.40 |
| Uncensored observations[2] | All data | 468 | 0 | 24.49 | 52.78 | 75.71 | 600.00 | 7.40 |
| | Cohort 1: Children | 99 | 0 | 15.12 | 28.47 | 35.20 | 204.57 | 7.42 |
| | Cohort 2: Adults | 369 | 0 | 29.92 | 59.29 | 82.11 | 600.00 | 7.40 |

N = total number per group; N cens = number of observations below the calculation limit (censored values); sd = standard deviation

[1]Two participants whose samples were repeatedly below the calculation limit in the anti-CdtB IgG assay were excluded from the analysis

[2]Robust regression on order statistics were used to calculate summary statistics, due to the high number of observations below the limit of calculation

´s rho = 0.72; $p<0.00001$) (Fig 2B) than among adults (Spearman´s rho = 0.35; $p<0.00001$) (Fig 2C). Conversely, the anti-Vi IgG titers did not exhibit a strong correlation with the anti-CdtB IgG titers (all data: Kendall´s tau = -0.03, $p = 0.14$; children: Kendall´s tau = -0.11, $p<0.0001$; adults: Kendall´s tau = 0.03, $p = 0.21$) or with anti-HlyE IgG titers (all data: Kendall

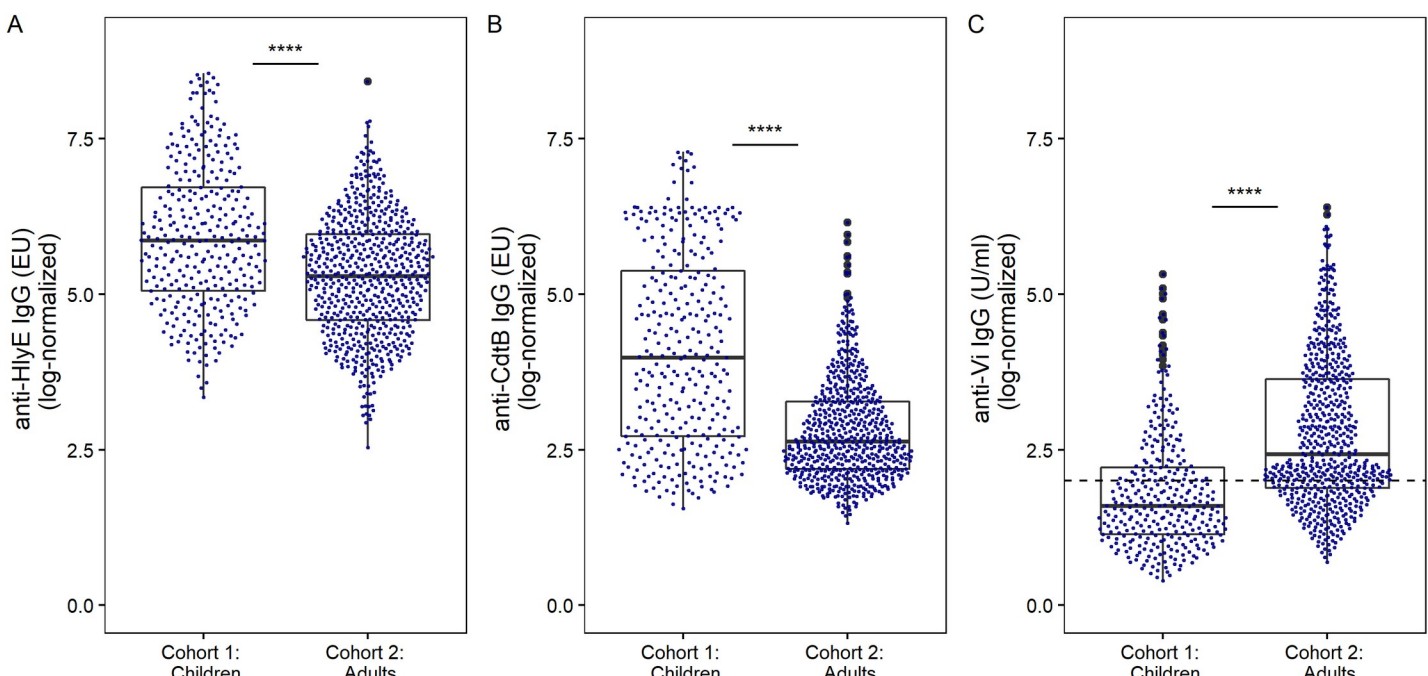

**Fig 1. The distribution of anti–S. Typhi serum IgG titers in children and adults in Lao PDR.** Each dot shows the antibody titer of an individual sample for (A) anti-HlyE IgG, (B) anti-CdtB IgG, and (C) anti-Vi IgG with an underlying boxplot. The dashed line in panel C represents the censoring limit, all data points below were treated as left-censored data. Differences between groups were assessed using Wilcoxon rank sum test followed by Dunn's post-hoc test with Bonferroni correction: ****$p<0.0001$.

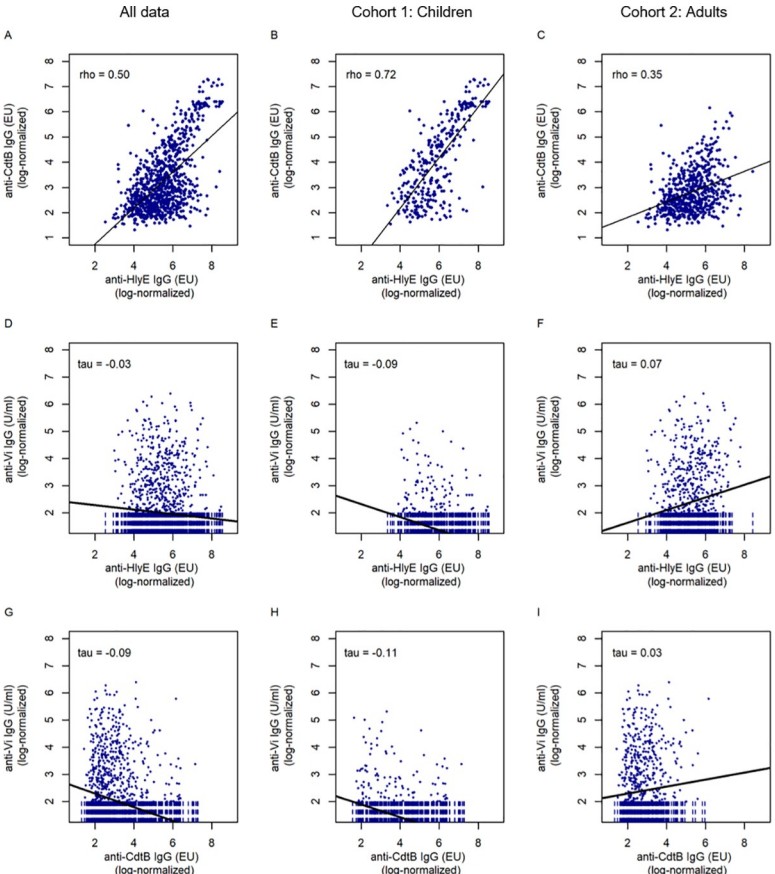

**Fig 2. Correlation of anti–*S.* Typhi serum IgG titers against three antigens in children and adults in Lao PDR.** A, B and C show the correlation between anti-HlyE IgG and anti-CdtB IgG for all data, cohort 1 and cohort 2 respectively. D, E and F show the correlation between anti-Vi IgG and anti-HlyE IgG for all data, cohort 1 and cohort 2 respectively. G, H and I show the correlation between anti-Vi IgG and anti-CdtB IgG for all data, cohort 1 and cohort 2 respectively. Black lines indicate predicted values from linear regression analysis in A, B and C and from Akritas-Thiel-Sen regression lines in D, E, F, G, H and I. Censored observations in D, E, F, G, H and I were plotted as vertical dashed lines.

´s tau = -0.03, *p* = 0.14; children: Kendall´s tau = -0.09, *p* = 0.003; adults: Kendall´s tau = 0.12, *p* = 0.004) (Fig 2).

## The relationship between age and anti–*S.* Typhi serum IgG antibodies

We fitted a linear model to investigate the relationship between anti-HlyE IgG titers and age. We found a significant negative relationship between anti-HlyE IgG titers and age (*p*<0.0001, adjusted $R^2$ = 3.4%). To assess the possibility of a non-linear relationship between age and anti-HlyE IgG titers, we fitted a generalized additive model (GAM) (Fig 3A). The fitted, non-linear trend in the GAM differed significantly (*p*<0.0001) from the linear trend fitted in the linear regression model (GAM: *p*<0.0001, adjusted $R^2$ = 15.5%, deviance explained = 16.2%). When comparing the overall model fit, the GAM demonstrated a better fit with the data than the linear model (GAM: AIC = 2649.85, BIC = 2698.65; linear model: AIC = 2767.8, BIC = 2782.3). Similarly, the GAM describing the relationship between anti-CdtB IgG titers and age (Fig 3B) was superior in terms of fit in comparison to the linear model (GAM: *p*<0.0001, adjusted $R^2$ = 30.4%, deviance explained = 30.9%, AIC = 2788.01, BIC = 2835.6;

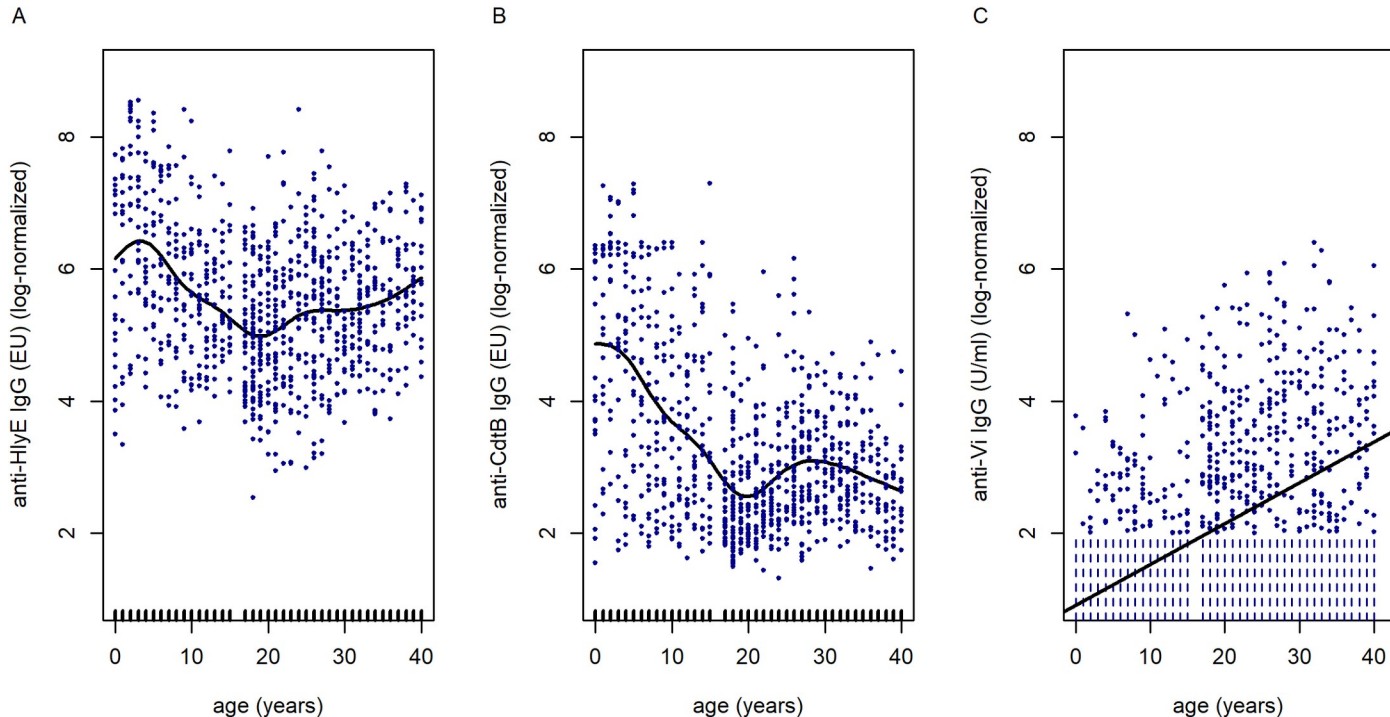

**Fig 3. Results of generalized additive and linear models assessing anti–*S*. Typhi IgG antibody prevalence in children and adults in Lao PDR as a function of age.** Non-linear smooths were fitted for age in the model for anti-HlyE IgG (A) and anti-CdtB IgG (B) data. The tick marks on the x-axis are observed data points. In panel C, the Akritas-Thiel-Sen regression line relating to the anti-Vi IgG titer data as function of age was plotted in order to account for the censored values (censored observations were plotted as vertical dashed lines). %dev. = the percent of the total model deviance explained.

linear model: $p < 0.0001$, adjusted $R^2$ = 17.4%, AIC = 2939.80, BIC = 2954.32) (Fig 3). The GAM for the anti-HlyE IgG and anti-CdtB IgG titers suggested the highest prevalence of antibody was in children aged <5 years (Figs 3A, 3B and S3). The fitted GAM trends identified a prominent decrease in antibody titers until the age of 20 years. S4 and S5 Figs show the GAM for anti-HlyE IgG and anti-CdtB IgG antibody titers as a function of age, by province in adults. We observed an upward trend of anti-HlyE IgG antibody titers with age in most provinces. For the anti-CdtB IgG data however, this trend was not consistent and differed by province. Lastly, the anti-Vi IgG data was fitted as a function of age, which showed a positive relationship using Akritas–Thiel–Sen non-parametric regression to account for the censored data (likelihood r = 0.33, $p < 0.0001$, Kendall's tau = 0.21; $p < 0.0001$) (Fig 3C); suggesting that anti-Vi IgG increases with age. In addition, we assessed the relationship between antibody titers and age in the Vientiane subset for which data from children and adults were available. A comparable pattern was observed: Anti-HlyE IgG and anti-CdtB IgG titers were highest in children and then decreased with age, while anti-Vi IgG titers were highest in adults (S6 Fig).

### Trends in anti–*S*. Typhi serum IgG antibodies regarding sex, occupation, and location

There was no significant difference for any of the anti-*S*. Typhi antibodies between male and female children or adults. We next investigated differences in anti-*S*. Typhi serum IgG antibodies according to occupation and study site in adults. A Kruskal Wallis test revealed a significant difference of anti-HlyE IgG among occupation groups ($p < 0.0001$) (S7 Fig). A *post-hoc* Dunn´s test with Bonferroni correction determined significant differences between the general

population with unspecified occupation ("other") (median anti-HlyE IgG = 312.4 EU) and students (median anti-HlyE IgG = 157.6 EU; $p<0.001$), between office workers (median anti-HlyE IgG = 259.2 EU) and students ($p<0.0001$), and between office workers and soldiers (median anti-HlyE IgG = 187.5 EU; $p<0.05$). Likewise, there was a significant difference between anti-CdtB IgG titers in students (median anti-CdtB IgG = 10.1 EU; $p<0.05$) and soldiers (median anti-CdtB IgG = 13.2 EU), the general population with unspecified occupation (median anti-CdtB IgG = 17.1 EU) and students ($p<0.01$) and between students and office workers (median anti-CdtB IgG = 15.2 EU; $p<0.001$). There was no significant difference in anti-Vi IgG titers between the occupation groups.

Further Kruskal Wallis tests revealed significant differences in anti-HlyE IgG titers, anti-CdtB IgG titers, and anti-Vi IgG titers between the different provinces ($p<0.0001$, $p<0.01$ and $p<0.05$ respectively) (Fig 4). A *post-hoc* Dunn´s test with Bonferroni correction identified a significant difference between several provinces. The distribution of anti-HlyE IgG titers differed between Khammuane (median anti-HlyE IgG = 255.9 EU) and Phongsaly (median anti-HlyE IgG = 144.0 EU; $p<0.05$), Attapeu (median anti-HlyE IgG = 142.3 EU; $p<0.01$) and Xayabouli (median anti-HlyE IgG = 157.8 EU; $p<0.01$), and between Vientiane (median anti-HlyE IgG titer = 272.0 EU; $p<0.01$) and Attapeu and Xayabouli ($p<0.05$) (Fig 4A). The median anti-CdtB IgG titres differed significantly between Khammouane (median anti-CdtB IgG = 15.3 EU) and Xayabouli (median anti-CdtB IgG = 9 EU; $p<0.05$) and between Vientiane (median anti-CdtB IgG = 16.1 EU) and Xayabouli ($p<0.01$) (Fig 4B). The only significant difference in median anti-Vi IgG titers was identified between Khammouane (median anti-Vi IgG = 20.3 U/ml) and Luang Prabang (median anti-Vi IgG = 7.8 U/ml; $p<0.05$) (Fig 4C).

## Discussion

The only reliable method for assessing the disease burden of typhoid fever is establishing population-based surveillance studies and introducing a standardized blood culture system. This approach recently uncovered a large, previously unobserved, burden of typhoid fever in sub-Saharan Africa [34]. These studies are complicated, expensive, and not sustainable outside of research funding. Additionally, clinical criteria, blood culture sensitivity, and site-specific nuances need to be taken into account throughout the study and can impact heavily on the corrected and uncorrected incidence figures. Serology may be a more scalable approach for typhoid fever surveillance. Serology is a fraction of the cost of population-based blood culture studies and could be used to highlight disease "hotspots" and identify which component of the population should be targeted for TCV introduction.

The antibody dynamics of active typhoid fever and carriage are yet to be fully characterized. However, both HlyE and CdtB have been shown to be informative *S*. Typhi antigens for serological investigations [15,16,35]. Anti-HlyE IgM, IgA and IgG responses are known to be elevated in confirmed typhoid fever cases in comparison to healthy controls [14]. Likewise, anti-CdtB IgM responses were higher in typhoid fever cases compared to controls using recombinant CdtB in an indirect ELISA [13]. As the principal component of typhoid fever vaccines, the Vi polysaccharide is the *S*. Typhi antigen most commonly used in serological studies [10,11,17].

We found that half of the participants had anti-Vi IgG antibody concentrations <7.4 U/ml, with an estimated median titer of 7.5 U/ml. Notably, the median baseline concentrations were higher in adults than children and we observed a positive association between anti-Vi IgG titer and age. These findings are largely consistent with previous reports [10,36]. The median anti-Vi titer after removing those below the calculation limit was 24.5 U/ml overall and 29.9 U/ml in adults, which was higher than previously reported median titers in healthy adults from the

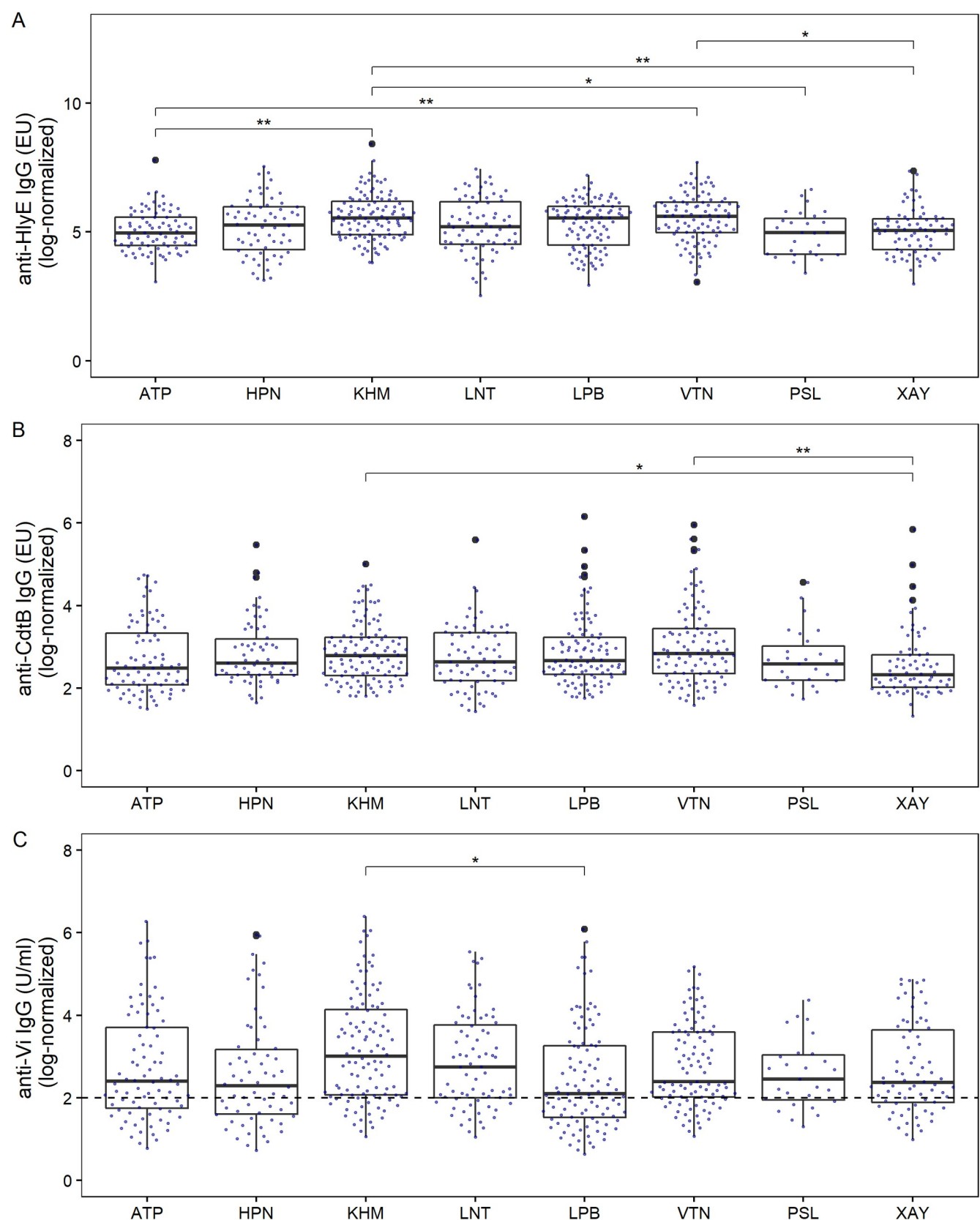

**Fig 4. The distribution of anti–S. Typhi serum IgG titers in adults in Lao PDR by province.** Each dot shows the measurement of an individual sample for (A) anti-HlyE IgG, (B) anti-CdtB IgG and (C) anti-Vi IgG with an underlying boxplot. Differences between groups were assessed using Kruskal-Wallis test followed by Dunn's post-hoc test with Bonferroni correction: $^*p<0.05$, $^{**}p<0.01$. ATP = Attapeu, HPN = Huaphan, KHM = Khammouane, LNT = Luang Namtha, LPB = Luang Prabang, VTN = Vientiane, PSL = Phongsaly, XAY = Xayabouli. The dashed line in panel C represents the censoring limit, all data points below were treated as left-censored data.

non-typhoid fever endemic countries Spain and Germany (8.6 U/ml and 21 U/ml, respectively) [37,38]. Conversely, the median anti-HlyE IgG and anti-CdtB IgG titers were substantially higher in children than in adults. The non-linear trend fitted to the HlyE IgG data identified a peak in antibody concentration in children below the age of 5 years; this peak was followed by a decrease and a secondary increase after the age of 20 years. The anti-CdtB IgG data followed largely the same trajectory; however, the increasing trend after the age of 20 years was not observed. The anti-HlyE IgG and anti-CdtB IgG titres correlated reasonably well with each other in the adult and children cohort. Notably, the anti-Vi IgG data, however, did not correlate well with either of the other two antibody profiles. A low correlation between the anti-Vi IgG titers and anti-HlyE/anti-CdtB IgG titers was surprising and could be due to different durability profiles. In contrast to the low correlation (tau -0.09; all data) between anti-CdtB IgG and anti-Vi IgG in our study, the IgM measurements against both of these antigens have demonstrated a moderate correlation (rho = 0.77) [7]. Data regarding the relationship between anti-S. Typhi antibody responses is currently lacking, indicating the need for further investigations.

Our data, measuring the presence of antibodies developed against the two protein antigens, anti-HlyE IgG and anti-CdtB IgG, potentially indicates greater S. Typhi exposure in children than adults. This contrasts a hospital based surveillance study in Vientiane, which reported no typhoid fever in children over the course of two years [22]. In a retrospective study in a central hospital in Vientiane, from 2000 to 2018, the median age of patients with confirmed typhoid fever was 21 years [20]. However, these hospital-based surveillance studies were performed in general hospitals, which have a low number of pediatric patients as they are admitted to more specialist facilities or have febrile illness managed in the community. Age-related Typhoid fever incidence patterns vary between high and low incidence settings: the incidence is typically highest in children in high incidence settings but more equally distributed in low incidence settings. In the global context, children are disproportionally affected by typhoid and paratyphoid fever with the highest incidence occurring in children aged between 5 and 9 years [39]. Serological testing is not a marker of active disease, but our data suggests children in the sampled locations in Lao PDR have been exposed to S. Typhi. A follow-up study specifically focusing on febrile disease and comparing blood culture data with serological testing in children in multiple provinces would allow a more accurate confirmation of the typhoid fever burden in children in Lao PDR.

Our study has limitations; the samples selected for this study were collected over different years and under the sampling framework of other studies. The data associated with the studies did not incorporate any health-related information including whether the participants had ever been diagnosed with typhoid fever. Serum samples from children were only available from two provinces in the Lao PDR. In Pongsaly province, fewer serum samples were available compared to the other locations, limiting our data analyses for this province. Additionally, the correlation between typhoid fever diseases and the antibody titer data was not yet established, complicating the interpretation of the titer data considerably. We currently have no cut-off for seroconversion with these antigens, so we used precise titers as a direct comparison between age groups and locations. Lastly, we cannot exclude the possibility of cross-reactive antibody responses, as HlyE and CdtB may be found in other bacteria [40,41].

In conclusion we identified a high prevalence of anti-HlyE and anti-CdtB IgG in the serum of children residing in Lao PDR. As these antigens are conserved within invasive *Salmonella* and known to stimulate an antibody response during infection our data suggest these may be tractable markers of disease exposure in typhoid fever endemic locations. Additional validation is required, but our approach is cost effective and scalable and may be developed to assess the burden of typhoid fever in countries.

## Supporting information

**S1 Fig. Map of study sites generating serum samples for anti-S. Typhi IgG serology in the Lao PDR.** PSL = Phongsaly, LNT = Luang Namtha, HPN = Huaphan, LPB = Luang Prabang, XAY = Xayabouli, VTN = Vientiane, BLX = Bolikhamxay, KHM = Khammouane, ATP = Attapeu. The map was created with QGIS (QGIS Development Team, 2018). The data regarding the administrative boundaries of Lao PDR were obtained from the Humanitarian Data Exchange website https://data.humdata.org/dataset/lao-admin-boundaries, dataset provided by the National Geographic Department of Lao PDR, 2019) and recreated under a CC BY-IGO license. Projection used: EPSG 4326 –WGS 84.
(TIF)

**S2 Fig. The distribution of anti–*S.* Typhi serum IgG titers in children and adults in Vientiane, Lao PDR.** Each dot shows the antibody titer of an individual sample for (A) anti-HlyE IgG, (B) anti-CdtB IgG, and (C) anti-Vi IgG with an underlying boxplot. The dashed line in panel C represents the censoring limit, all data points below were treated as left-censored data. Differences between groups were assessed using Wilcoxon rank sum test followed by Dunn's post-hoc test with Bonferroni correction: ***$p<0.001$, ****$p<0.0001$.
(TIF)

**S3 Fig. Results of generalized additive models assessing anti–*S.* Typhi IgG antibody prevalence in children and adults in Lao PDR as a function of birth year.** Non-linear smooths were fitted for birth year in the model for anti-HlyE IgG (A) and anti-CdtB IgG (B) data. Shaded bands represent the pointwise 95%-confidence interval.
(TIF)

**S4 Fig. Results of the generalized additive model assessing anti-HlyE IgG titer in adults in Lao PDR as a function of age by province.** Shaded bands represent the pointwise 95%-confidence interval. ATP = Attapeu, HPN = Huaphan, KHM = Khammouane, LNT = Luang Namtha, LPB = Luang Prabang, VTN = Vientiane, PSL = Phongsaly, XAY = Xayabouli.
(TIF)

**S5 Fig. Results of the generalized additive model assessing anti-CdtB IgG titer in adults in Lao PDR a function of age by province.** Shaded bands represent the pointwise 95%-confidence interval. ATP = Attapeu, HPN = Huaphan, KHM = Khammouane, LNT = Luang Namtha, LPB = Luang Prabang, VTN = Vientiane, PSL = Phongsaly, XAY = Xayabouli.
(TIF)

**S6 Fig. Results of generalized additive and linear models assessing anti–S. Typhi IgG antibody prevalence in children and adults in Vientiane, Lao PDR as a function of age.** Non-linear smooths were fitted for age in the model for anti-HlyE IgG (A) and anti-CdtB IgG (B) data. The tick marks on the x-axis are observed data points. In panel C, the Akritas-Thiel-Sen regression line relating to the anti-Vi IgG titer data as function of age was plotted in order to account for the censored values (censored observations were plotted as vertical dashed lines).

%dev. = the percent of the total model deviance explained.
(TIF)

**S7 Fig. The distribution of anti–*S*. Typhi serum IgG titers in adults in Lao PDR by occupa-
tion.** Each dot shows the measurement of an individual sample for (A) anti-HlyE IgG, (B)
anti-CdtB IgG and (C) anti-Vi IgG with an underlying boxplot. Differences between groups
were assessed using Kruskal-Wallis test followed by Dunn's post-hoc test with Bonferroni cor-
rection: $^*p<0.05$, $^{**}p<0.01$, $^{***}p<0.001$, $^{****}p<0.0001$. If not specified otherwise, differences
in titer data were non-significant. Participants whose occupation is not specified are grouped
into "other". The dashed line in panel C represents the censoring limit, all data points below
were treated as left-censored data.
(TIF)

## Acknowledgments

We wish to acknowledge investigators, especially Vilaysone Khounvisith and Phonethipsavanh
Nouanthong, from the named studies for providing access to the samples and data of the study
participants. We are thankful to our collaborators at Luxembourg Development Cooperation,
the Lao Red Cross and Provincial and district hospitals and health centres. Latdavone Khenka
and Bounta Vongphachanh helped with the experiments. We wish to acknowledge Dr. Paul
Brey and staff at the Institut Pasteur du Laos for their support.

## Author Contributions

**Conceptualization:** Lisa Hefele, Antony P. Black, Stephen Baker.

**Formal analysis:** Lisa Hefele.

**Funding acquisition:** Lisa Hefele, Antony P. Black, Stephen Baker.

**Investigation:** Lisa Hefele, Claude P. Muller, Paula Russell.

**Methodology:** Lisa Hefele, Trinh Van Tan, Nguyen Tri Minh, Nguyen Duc Hoang, Siriphone
Virachith, Claude P. Muller, Josefin Bartholdson Scott, Chau Nguyen Ngoc Minh, Tran Vu
Thieu Nga.

**Project administration:** Lisa Hefele, Stephen Baker.

**Resources:** Antony P. Black, Trinh Van Tan, Nguyen Tri Minh, Nguyen Duc Hoang, Siri-
phone Virachith, Judith M. Hübschen, Paula Russell, Josefin Bartholdson Scott, Chau
Nguyen Ngoc Minh, Tran Vu Thieu Nga.

**Supervision:** Antony P. Black, Nguyen Duc Hoang, Claude P. Muller, Judith M. Hübschen,
Stephen Baker.

**Writing – original draft:** Lisa Hefele, Stephen Baker.

**Writing – review & editing:** Antony P. Black, Stephen Baker.

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
