## [Decision Letter · Decision Letter 0]

16 Jun 2021

Dear Dr. Baker,

Thank you very much for submitting your manuscript "An age-stratified serosurvey against purified Salmonella enterica serovar Typhi antigens in the Lao People´s Democratic Republic" for consideration at PLOS Neglected Tropical Diseases. As with all papers reviewed by the journal, your manuscript was reviewed by members of the editorial board and by several independent reviewers. In light of the reviews (below this email), we would like to invite the resubmission of a significantly-revised version that takes into account the reviewers' comments. 

We cannot make any decision about publication until we have seen the revised manuscript and your response to the reviewers' comments. Your revised manuscript is also likely to be sent to reviewers for further evaluation.

Sincerely,

Travis J Bourret

Associate Editor

Sujay Chattopadhyay

Deputy Editor

Reviewer's Responses to Questions

**Key Review Criteria Required for Acceptance?**

**Methods**

-Are the objectives of the study clearly articulated with a clear testable hypothesis stated?

-Is the study design appropriate to address the stated objectives?

-Is the population clearly described and appropriate for the hypothesis being tested?

-Is the sample size sufficient to ensure adequate power to address the hypothesis being tested?

-Were correct statistical analysis used to support conclusions?

-Are there concerns about ethical or regulatory requirements being met?

Reviewer #1: The objectives as set out are noble and would answer key research questions in this area.

The study design should have had an element of longitudinal collection of serological data on the study population to understand how long the markers of typhoid exposure persist in the individuals and how this relates to patient presentation or carriage status

Sample size and analytical framework are well designed, and no concerns on ethics

Reviewer #2: The methodology of the study is clearly presented and articulated.

However, the specificity of the 3 antibodies for S. Typhi (vs other non-typhoidal serovars for instance), and their detectability in the serum are not compared or discussed. This absence limits the conclusion of the study but could be included as an additional experiment testing the presence of these antibodies in the blood of patients after typhoid fever was detected (for this purpose, the sample collection could be unrestrained to Lao if necessary). Alternatively, references to previously published work providing this comparison (if existing) should be included and discussed.

Minor point:

- In the “Study population” subsection of the Methods, the author should mention that the populations included in the study have not received the Vi polysaccharide-based vaccine.

- The authors may comment if both cohorts (adult and children) had similar living conditions (access to running water, etc).

Reviewer #3: 1. I realize that these blood samples were from other studies, but is there any way to determine if any samples were from people vaccinated for typhoid fever? If not, could this be stated in the manuscript.

2. The adult blood donor age range from 17 – 40 years seems a little bit narrow. How were the individuals randomly selected? What was the overall range of the 5,018 samples that the 640 adult samples were randomly selected from for the study?

3. Could more details be provided about the commercial anti-Vi IgG ELISA kit, such as were the manufacturer’s instructions followed exactly without any modifications. If modifications, what were those modifications? Wavelength? As this is a critical method of the manuscript providing the reader with enough detail to potentially replicate it is vital.

4. Why was HlyE IgG ELISA included in the study? The reference cited for the in-house developed ELISA does not appear to include hlyE gene, therefore what is the specificity and sensitivity of this ELISA?

**Results**

-Does the analysis presented match the analysis plan?

-Are the results clearly and completely presented?

-Are the figures (Tables, Images) of sufficient quality for clarity?

Reviewer #1: Analysis as presented is well thought out and clearly done including Tables and Figures.

Major concerns:

Page 8, Line 207: We observed an upward trend of antibody titers with age in some provinces; however, this trend t was not consistent and differed by province. Lastly, the anti-Vi IgG data was fitted as a function of age, which showed a positive relationship using Akritas–Theil–Sen non-parametric regression to account for the censored data (likelihood r = 0.33, p<0.0001, Kendall`s tau = 0.21; p<0.0001) (Figure 3); suggesting that anti-Vi IgG increases with age.

Comment: what is the context of these titres in relation to active disease and carriage in endemic settings? Granted that they provide guidance on exposure, they are a good indicator and guide for deployment of vaccines to reduce carriage and transmission in the community.

Can they be used for rapid detection of disease? 

Page 9, Line 232: The distribution of anti-HlyE IgG titers differed between Khammuane (median anti-HlyE IgG = 255.9 EU) and Phongsaly (median anti-HlyE IgG = 144.0 EU),Attapeu (median anti-HlyE IgG = 142.3 EU) and Xayabouli (median anti-HlyE IgG = 157.8 EU), and between Vientiane (median anti-HlyE IgG titer = 272.0 EU) and Attapeu and Xayabouli (Figure 4A).The median anti-CdtB IgG titres differed significantly between Khammouane (median anti-CdtB IgG= 15.3 EU) and Xayabouli (median anti-CdtB IgG = 9 EU) and between Vientiane (median anti-CdtB IgG = 16.1 EU) and Xayabouli (Figure 4B). The only significant difference in median anti-Vi IgG titers was identified between Khammouane (median anti-Vi IgG = 20.3 U/ml) and Luang Prabang (median anti-Vi IgG = 7.8 U/ml) (Figure 4C).

Comment: what is the reason for the variation according to geographic distribution?

Any specific risk attributes unique to the specific study sites?

Reviewer #2: In general, the results are presented in a very clear and exhaustive manner.

Minor point:

- Line 207-208 (referring to Figures S3-S4): the authors wrote “We observed an upward trend of antibody titers with age in some provinces; however, this trend t was not consistent and differed by province.“ This statement should be detailed/refined. For instance, the trend seems consistent between provinces for anti-HyE IgG but not for anti-CdtB IgG. Was the population size of some provinces limiting the analysis? If yes, this should be mentioned in the text.

Reviewer #3: 5. Lines 213-225, it would be good to put in the p-Values for those significant differences.

**Conclusions**

-Are the conclusions supported by the data presented?

-Are the limitations of analysis clearly described?

-Do the authors discuss how these data can be helpful to advance our understanding of the topic under study?

-Is public health relevance addressed?

Reviewer #1: Page 10, Line 252: We found that half of the participants had anti-Vi IgG antibody concentrations <7.4 U/ml, with an estimated median titer of 7.5 U/ml.

Comment: How do these titres correlate with actual typhoid disease?

Page 11, Line 276: A follow up study specifically focusing on febrile disease and comparing blood culture data with serological testing in children in multiple provinces would allow a more accurate confirmation of the typhoid burden in children in Lao PDR.

Comment: This correlation of active disease and serology is crucial to provide a diagnostic basis for use of serological testing suggested for diagnostic purposes. 

At the moment, the investigators should cite lack of this correlation as a major limitation in interpretation of their - one cannot tell if the raised levels of antigens is marker for low-grade exposure, active disease or passive carriage

Reviewer #2: While the data obtained are exhaustively analyzed and described, the interpretation of the results remains superficial. The study – in particular, the discussion section - would highly beneficiate from a more in-depth biological contextualization and a consideration of the results in a broader context.

Here are several elements that may be further discussed:

(1) Differences observed between antibodies:

- The 3 different IgG should be further introduced. E.g., CdtB stands for Cytolethal Distending Toxin B, and HlyE stands for Haemolysin E. Both are bacteria toxins produced by several bacteria species. The specificity of these antigens to S. Typhi should be introduced (see also “methods” comments).

- A key result of the study is the absence of correlation between the anti-Vi polysaccharide IgG titers and the anti-toxin (CdtB /HlyE) IgG titers, and their different distribution in the age-stratified population. This result was unexpected and the authors should provide some hypotheses in the discussion. Could this be explained by different duration of the seroconversions? By the specificity or detectability of the antibodies? (see also “methods” comments).

- As one purpose of the study is to orientate future survey strategies and ease the introduction of a vaccination program, the authors may provide more direct guidance on which antibody detection should be favored for future serological surveys, and possibly comment on the availability of the different antibodies and their production cost.

(2) Difference observed between age, location, occupation:

- How could the difference of serology depending on the location and occupation be interpreted? Could it be linked with different access to running water, food origin, etc? Could the occupation be commented on in terms of Salmonella exposure? Could the province be commented on in terms of previous outbreak locations?

- As the authors checked both the serology as a function of the age and as a function of the year of birth, it would be interesting to discuss if the antibody distribution in the age-stratified population may result from a different exposure depending on the year (considering the date of S. Typhi outbreaks) or from the timing of biological processes.

 - Can these results be put in context with the situation in neighboring countries with a high prevalence of S. Typhi?

Reviewer #3: 6. The authors do not discuss why there is no correlation between anti-Vi titers and anti-CdtB or anti-HlyE titers, it seems if this is a critical issue that should be discussed by the authors. The authors discuss the differences with age, but if all of these are signs of previous typhoid fever infection, would you not expect a correlation between the major polysaccharide antigen and potential toxins. Additionally, why do you think there is an age difference in the antigens that are inducing an immune response? Many enteric pathogens produce cytolethal distending toxin (CDT) is it possible that this is due to a cross-reactivity or response to other enteric pathogens more common in children such as Campylobacter. All of these should be discussed for the paper.

7. Is it possible that the children’s samples only coming from two provinces, which were significantly higher for anti-CdtB or anti-HlyE titers could be biasing some of the results for the different provinces? This should be addressed by the authors and how this bias was avoided.

**Editorial and Data Presentation Modifications?**

Reviewer #1: (No Response)

Reviewer #2: Minor points on the text:

- In the introduction, lines 61-62, the authors wrote: “A lack of longitudinal incidence data in many countries where typhoid is suspected to be endemic is a major barrier for the introduction of typhoid conjugate vaccines (TCVs)” I suggest that the authors add a sentence explicitly explaining why this constitutes a major barrier for TCV introduction.

- Direct references to given subfigure panels (for instance “Figure 1A”) are often missing in the text. It would help the readability of the results to specify systematically which part of the figure the text is referring to.

- Regarding the availability of data and materials, the authors state “All data generated or analyzed during this study are included in this published article and its supplementary information files.” Yet, it is unclear how to get access to the full raw anonymized dataset of the epidemiological study. As such data could be directly used for other investigations, it would be critical to provide this information.

Minor points on the Figures:

- Statistical information should be added in Figure 3.

- “ns” for “non-significant” could be added in Figure 5C.

Reviewer #3: 1. The authors switch between using “typhoid fever” and “typhoid” randomly throughout the manuscript. One term should be used consistently throughout the manuscript to avoid reader confusion.

2. The reference section should have genus and species names italicized for proper formatting.

**Summary and General Comments**

Reviewer #1: This is a manuscript that provides data on serological surveillance for typhoid specific antigen markers that maybe useful for assessment of exposure to typhoid in disease endemic settings. What is not clear is how long these markers persist in the individuals in the community and are they specific in magnitude/persistence in acute disease compared to carriage? It would have made more sense to have some longitudinal data to assess and answer this question.

Reviewer #2: The study systematically compares the presence of 3 anti–S. Typhi IgG antibodies in the blood of Laotian of different age, sex, location, and occupation. The outcome of the study could be a cornerstone for the development of a vaccination strategy. 

Strong points: The investigation is performed properly, the manuscript is presented clearly and the conclusions well justified. Due to the low health surveillance in Lao, the collection of this data is a tour de force and offers very high public health relevance.

Weak points: The manuscript currently lacks biological contextualization of the investigation and discussion of the results in a broader context (see more detailed comments in “conclusions”).

Reviewer #3: (No Response)

PLOS authors have the option to publish the peer review history of their article (what does this mean?). If published, this will include your full peer review and any attached files.

Reviewer #1: No

Reviewer #2: No

Reviewer #3: No
---

## [Decision Letter · Decision Letter 1]

15 Sep 2021

Dear Dr. Baker,

Thank you very much for submitting your manuscript "An age-stratified serosurvey against purified Salmonella enterica serovar Typhi antigens in the Lao People´s Democratic Republic" for consideration at PLOS Neglected Tropical Diseases. As with all papers reviewed by the journal, your manuscript was reviewed by members of the editorial board and by several independent reviewers. The reviewers appreciated the attention to an important topic. Based on the reviews, we are likely to accept this manuscript for publication, providing that you modify the manuscript according to the review recommendations. 

Sincerely,

Travis J Bourret

Associate Editor

Sujay Chattopadhyay, PhD

Deputy Editor

Reviewer's Responses to Questions

**Key Review Criteria Required for Acceptance?**

**Methods**

-Are the objectives of the study clearly articulated with a clear testable hypothesis stated?

-Is the study design appropriate to address the stated objectives?

-Is the population clearly described and appropriate for the hypothesis being tested?

-Is the sample size sufficient to ensure adequate power to address the hypothesis being tested?

-Were correct statistical analysis used to support conclusions?

-Are there concerns about ethical or regulatory requirements being met?

Reviewer #2: The authors have provided satisfying answers and text edits to my methods-related comments.

Reviewer #3: Methods are fine and clearly stated.

Reviewer #4: The objectives of the study are clearly spelled out. The study design is good; the population is clearly described and a large enough size of study participants are recruited to make the results statistically meaningful. There are no questions regarding ethical or regulatory requirements.

**Results**

-Does the analysis presented match the analysis plan?

-Are the results clearly and completely presented?

-Are the figures (Tables, Images) of sufficient quality for clarity?

Reviewer #2: The authors have provided satisfying answers and text edits to my results-related comments.

Reviewer #3: There are no issues with the results.

Reviewer #4: The results are clearly presented and the data support the conclusions. Figures and tables are clear.

**Conclusions**

-Are the conclusions supported by the data presented?

-Are the limitations of analysis clearly described?

-Do the authors discuss how these data can be helpful to advance our understanding of the topic under study?

-Is public health relevance addressed?

Reviewer #2: The authors have replied to all the points previously raised in their point-by-point answer. However, in most cases, they could not directly improve the results and conclusions within the manuscript, but rather detailed in their rebuttal the current limitations that restricted the study (lack of current knowledge on the antibody responses, lack of epidemiological information in Lao, etc.). Given these circumstances, the authors pushed the analysis and interpretation to their best extend and avoided drawing conclusions or interpretations that could not be directly supported by the data.

Reviewer #3: Conclusions from the study are supported by the data from the study, but a little more in depth thought as to the meaning of the results would help to strengthen the manuscript.

Reviewer #4: Conclusions are borne out by the data. Limitations of the study are also addressed by the authors.

**Editorial and Data Presentation Modifications?**

Reviewer #2: The authors have considered and implemented my editorial suggestions.

Reviewer #3: (No Response)

Reviewer #4: (None)

**Summary and General Comments**

Reviewer #2: While the limited conclusions of the study may impair its visibility, it also highlights the necessity to support and develop similar studies until reaching a broader understanding of typhoid epidemiology. However, I regret that the authors were unable to consequently revised the text to contextualize their study as suggested. This would have made the manuscript more attractive for a broader audience (only minor edits were implemented in this regard despite the editor's recommendation to significantly revise the manuscript). In its current state, as the manuscript mainly displays epidemiological observations but no or limited interpretations of the results, it may fit best a journal dedicated to epidemiological reports.

Reviewer #3: Overall, I think the manuscript has been greatly improved, but there are still a couple of minor issues that should be resolved prior to publication:

As the central finding of the study is that children have higher titers for HlyE and CdtB compared to adults, however this includes adults from numerous provinces and children from only two provinces. It would be interesting to see if this continues to be true by narrowing the analysis down to just the two provinces that all the children samples came from for the study (e.g. Vientiane & Bolikhamxay). 

I would like to see them expand the thoughts on the low correlation between anti-Vi titers and anti-HlyE/anti-CdtB titers, as this seems critical to the study, but the authors do not offer a very detailed explanation other than different durability profiles. Is there anything in the literature to support this thought? Anything in the literature to offer alternative ideas?

For the difference in children versus adults that is opposite from the hospital based study in Vientiane, what do other typhoid fever studies in other countries suggest about the results? Have other studies found differences between serum based versus hospital based studies in other countries? 

Overall, I think a little bit more thought on the meaning of the results is needed for the discussion prior to publication. Otherwise it is a solid paper.

Reviewer #4: (No Response)

PLOS authors have the option to publish the peer review history of their article (what does this mean?). If published, this will include your full peer review and any attached files.

Reviewer #2: No

Reviewer #3: No

Reviewer #4: No

Figure Files:

Data Requirements:

Reproducibility:

References

---

## [Decision Letter · Decision Letter 2]

22 Nov 2021

Dear Dr. Baker,

We are pleased to inform you that your manuscript 'An age-stratified serosurvey against purified Salmonella enterica serovar Typhi antigens in the Lao People´s Democratic Republic' has been provisionally accepted for publication in PLOS Neglected Tropical Diseases.

Best regards,

Travis J Bourret

Associate Editor

Sujay Chattopadhyay

Deputy Editor

Reviewer's Responses to Questions

**Key Review Criteria Required for Acceptance?**

**Methods**

-Are the objectives of the study clearly articulated with a clear testable hypothesis stated?

-Is the study design appropriate to address the stated objectives?

-Is the population clearly described and appropriate for the hypothesis being tested?

-Is the sample size sufficient to ensure adequate power to address the hypothesis being tested?

-Were correct statistical analysis used to support conclusions?

-Are there concerns about ethical or regulatory requirements being met?

Reviewer #2: (No Response)

Reviewer #4: The methodology is adequately described and backed by literature references. The plots neatly describe the data. This together with the large study size allows the authors to draw convincing and useful conclusions.

**Results**

-Does the analysis presented match the analysis plan?

-Are the results clearly and completely presented?

-Are the figures (Tables, Images) of sufficient quality for clarity?

Reviewer #2: (No Response)

Reviewer #4: The results clearly show, as reported, that antibody responses against typhoid fever antigens HlyE and CdtB are notably higher in children than in adults. However, when using the bacterial polysaccharide antigen Vi, antibody responses are higher in adults than in children.

**Conclusions**

-Are the conclusions supported by the data presented?

-Are the limitations of analysis clearly described?

-Do the authors discuss how these data can be helpful to advance our understanding of the topic under study?

-Is public health relevance addressed?

Reviewer #2: (No Response)

Reviewer #4: The conclusions are supported by the data.

The limitations are addressed and described. The authors discuss how their data will further our understanding.

The study is of great public health relevance and is discussed.

**Editorial and Data Presentation Modifications?**

Reviewer #2: (No Response)

Reviewer #4: Accept

**Summary and General Comments**

Reviewer #2: The authors have thoughtfully considered all my comments and edited the manuscript accordingly. I don't have further requests. Therefore, I support the manuscript for its publication in PNTD.

Reviewer #4: There are no remaining issues.

PLOS authors have the option to publish the peer review history of their article (what does this mean?). If published, this will include your full peer review and any attached files.

Reviewer #2: **Yes: **Virginie Stévenin

Reviewer #4: No

---

## [Editor Report · Acceptance letter]

7 Dec 2021

Dear Professor Baker,

We are delighted to inform you that your manuscript, "An age-stratified serosurvey against purified *Salmonella enterica* serovar Typhi antigens in the Lao People´s Democratic Republic," has been formally accepted for publication in PLOS Neglected Tropical Diseases.

Best regards,

Shaden Kamhawi

co-Editor-in-Chief

Paul Brindley

co-Editor-in-Chief
